# PROVABLE ROBUSTNESS AGAINST ALL ADVERSARIAL $l_p$-PERTURBATIONS FOR $p \geq 1$

**Francesco Croce**
University of Tübingen, Germany

**Matthias Hein**
University of Tübingen, Germany

## ABSTRACT

In recent years several adversarial attacks and defenses have been proposed. Often seemingly robust models turn out to be non-robust when more sophisticated attacks are used. One way out of this dilemma are provable robustness guarantees. While provably robust models for specific $l_p$-perturbation models have been developed, we show that they do not come with any guarantee against other $l_q$-perturbations. We propose a new regularization scheme, MMR-Universal, for ReLU networks which enforces robustness wrt $l_1$- *and* $l_\infty$-perturbations and show how that leads to the first provably robust models wrt any $l_p$-norm for $p \geq 1$.

## 1 INTRODUCTION

The vulnerability of neural networks against adversarial manipulations (Szegedy et al., 2014; Goodfellow et al., 2015) is a problem for their deployment in safety critical systems such as autonomous driving and medical applications. In fact, small perturbations of the input which appear irrelevant or are even imperceivable to humans change the decisions of neural networks. This questions their reliability and makes them a target of adversarial attacks.

To mitigate the non-robustness of neural networks many empirical defenses have been proposed, e.g. by Gu & Rigazio (2015); Zheng et al. (2016); Papernot et al. (2016); Huang et al. (2016); Bastani et al. (2016); Madry et al. (2018), but at the same time more sophisticated attacks have proven these defenses to be ineffective (Carlini & Wagner, 2017; Athalye et al., 2018; Mosbach et al., 2018), with the exception of the adversarial training of Madry et al. (2018). However, even these $l_\infty$-adversarially trained models are not more robust than normal ones when attacked with perturbations of small $l_p$-norms with $p \neq \infty$ (Sharma & Chen, 2019; Schott et al., 2019; Croce et al., 2019b; Kang et al., 2019). The situation becomes even more complicated if one extends the attack models beyond $l_p$-balls to other sets of perturbations (Brown et al., 2017; Engstrom et al., 2017; Hendrycks & Dietterich, 2019; Geirhos et al., 2019).

Another approach, which fixes the problem of overestimating the robustness of a model, is provable guarantees, which means that one certifies that the decision of the network does not change in a certain $l_p$-ball around the target point. Along this line, current state-of-the-art methods compute either the norm of the minimal perturbation changing the decision at a point (e.g. Katz et al. (2017); Tjeng et al. (2019)) or lower bounds on it (Hein & Andriushchenko, 2017; Raghunathan et al., 2018; Wong & Kolter, 2018). Several new training schemes like (Hein & Andriushchenko, 2017; Raghunathan et al., 2018; Wong & Kolter, 2018; Mirman et al., 2018; Croce et al., 2019a; Xiao et al., 2019; Gowal et al., 2018) aim at both enhancing the robustness of networks and producing models more amenable to verification techniques. However, all of them are only able to prove robustness against a single kind of perturbations, typically either $l_2$- or $l_\infty$-bounded, and not wrt all the $l_p$-norms simultaneously, as shown in Section 5. Some are also designed to work for a specific $p$ (Mirman et al., 2018; Gowal et al., 2018), and it is not clear if they can be extended to other norms.

The only two papers which have shown, with some limitations, non-trivial empirical robustness against multiple types of adversarial examples are Schott et al. (2019) and Tramèr & Boneh

(2019), which resist to $l_0$- resp. $l_1$-, $l_2$- and $l_\infty$-attacks. However, they come without provable guarantees and Schott et al. (2019) is restricted to MNIST.

In this paper we aim at robustness against all the $l_p$-bounded attacks for $p \geq 1$. We study the non-trivial case where none of the $l_p$-balls is contained in another. If $\epsilon_p$ is the radius of the $l_p$-ball for which we want to be provably robust, this requires: $d^{\frac{1}{p} - \frac{1}{q}} \epsilon_q > \epsilon_p > \epsilon_q$ for $p < q$ and $d$ being the input dimension. We show that, for normally trained models, for the $l_1$- and $l_\infty$-balls we use in the experiments none of the adversarial examples constrained to be in the $l_1$-ball (i.e. results of an $l_1$-attack) belongs to the $l_\infty$-ball, and vice versa. This shows that certifying the *union* of such balls is significantly more complicated than getting robust in only one of them, as in the case of the union the attackers have a much larger variety of manipulations available to fool the classifier.

We propose a technique which allows to train piecewise affine models (like ReLU networks) which are *simultaneously provably robust to all the $l_p$-norms* with $p \in [1, \infty]$. First, we show that having guarantees on the $l_1$- and $l_\infty$-distance to the decision boundary and region boundaries (the borders of the polytopes where the classifier is affine) is sufficient to derive meaningful certificates on the robustness wrt all $l_p$-norms for $p \in (1, \infty)$. In particular, our guarantees are independent of the dimension of the input space and thus go beyond a naive approach where one just exploits that all $l_p$-metrics can be upper- and lower-bounded wrt any other $l_q$-metric. Then, we extend the regularizer introduced in Croce et al. (2019a) so that we can directly maximize these bounds at training time. Finally, we show the effectiveness of our technique with experiments on four datasets, where the networks trained with our method are the first ones having non-trivial provable robustness wrt $l_1$-, $l_2$- and $l_\infty$-perturbations.

## 2   Local properties and robustness guarantees of ReLU networks

It is well known that feedforward neural networks (fully connected, CNNs, residual networks, DenseNets etc.) with piecewise affine activation functions, e.g. ReLU, leaky ReLU, yield continuous piecewise affine functions (see e.g. Arora et al. (2018); Croce & Hein (2018)). Croce et al. (2019a) exploit this property to derive bounds on the robustness of such networks against adversarial manipulations. In the following we recall the guarantees of Croce et al. (2019a) wrt a single $l_p$-perturbation which we extend in this paper to simultaneous guarantees wrt all the $l_p$-perturbations for $p$ in $[1, \infty]$.

### 2.1   ReLU networks as piecewise affine functions

Let $f : \mathbb{R}^d \to \mathbb{R}^K$ be a classifier with $d$ being the dimension of the input space and $K$ the number of classes. The classifier decision at a point $x$ is given by $\arg\max_{r=1,...,K} f_r(x)$. In this paper we deal with ReLU networks, that is with ReLU activation function (in fact our approach can be easily extended to any piecewise affine activation function e.g. leaky ReLU or other forms of layers leading to a piecewise affine classifier as in Croce et al. (2019b)).

**Definition 2.1** *A function $f : \mathbb{R}^d \to \mathbb{R}$ is called* piecewise affine *if there exists a finite set of polytopes $\{Q_r\}_{r=1}^M$ (referred to as* linear regions *of $f$) such that $\cup_{r=1}^M Q_r = \mathbb{R}^d$ and $f$ is an affine function when restricted to every $Q_r$.*

Denoting the activation function as $\sigma$ ($\sigma(t) = \max\{0, t\}$ if ReLU is used) and assuming $L$ hidden layers, we have the usual recursive definition of $f$ as

$$g^{(l)}(x) = W^{(l)} f^{(l-1)}(x) + b^{(l)}, \quad f^{(l)}(x) = \sigma(g^{(l)}(x)), \quad l = 1, \ldots, L,$$

with $f^{(0)}(x) \equiv x$ and $f(x) = W^{(L+1)} f^{(L)}(x) + b^{(L+1)}$ the output of $f$. Moreover, $W^{(l)} \in \mathbb{R}^{n_l \times n_{l-1}}$ and $b^{(l)} \in \mathbb{R}^{n_l}$, where $n_l$ is the number of units in the $l$-th layer ($n_0 = d$, $n_{L+1} = K$).

For the convenience of the reader we summarize from Croce & Hein (2018) the description of the polytope $Q(x)$ containing $x$ and affine form of the classifier $f$ when restricted to $Q(x)$. We assume that $x$ does not lie on the boundary between polytopes (this is almost always

true as faces shared between polytopes are of lower dimension). Let $\Delta^{(l)}, \Sigma^{(l)} \in \mathbb{R}^{n_l \times n_l}$ for $l = 1, \ldots, L$ be diagonal matrices defined elementwise as

$$\Delta^{(l)}(x)_{ij} = \begin{cases} \text{sign}(f_i^{(l)}(x)) & \text{if } i = j, \\ 0 & \text{else.} \end{cases}, \qquad \Sigma^{(l)}(x)_{ij} = \begin{cases} 1 & \text{if } i = j \text{ and } f_i^{(l)}(x) > 0, \\ 0 & \text{else.} \end{cases}.$$

This allows us to write $f^{(l)}(x)$ as composition of affine functions, that is

$$f^{(l)}(x) = W^{(l)}\Sigma^{(l-1)}(x)\Big(W^{(l-1)}\Sigma^{(l-2)}(x) \times \Big(\ldots\Big(W^{(1)}x + b^{(1)}\Big)\ldots\Big) + b^{(l-1)}\Big) + b^{(l)},$$

which we simplify as $f^{(l)}(x) = V^{(l)}x + a^{(l)}$, with $V^{(l)} \in \mathbb{R}^{n_l \times d}$ and $a^{(l)} \in \mathbb{R}^{n_l}$ given by

$$V^{(l)} = W^{(l)}\Big(\prod_{j=1}^{l-1} \Sigma^{(l-j)}(x)W^{(l-j)}\Big) \text{ and } a^{(l)} = b^{(l)} + \sum_{j=1}^{l-1}\Big(\prod_{m=1}^{l-j} W^{(l+1-m)}\Sigma^{(l-m)}(x)\Big)b^{(j)}.$$

A forward pass through the network is sufficient to compute $V^{(l)}$ and $b^{(l)}$ for every $l$. The polytope $Q(x)$ is given as intersection of $N = \sum_{l=1}^{L} n_l$ half spaces defined by

$$Q(x) = \bigcap_{l=1,\ldots,L}\bigcap_{i=1,\ldots,n_l} \big\{z \in \mathbb{R}^d \,\big|\, \Delta^{(l)}(x)_{ii}\big(V_i^{(l)}z + a_i^{(l)}\big) \geq 0\big\},$$

Finally, the affine restriction of $f$ to $Q(x)$ is $f(z)|_{Q(x)} = f^{(L+1)}|_{Q(x)}(z) = V^{(L+1)}z + a^{(L+1)}$.

Let $q$ be defined via $\frac{1}{p} + \frac{1}{q} = 1$ and $c$ the correct class of $x$. We introduce

$$d_{p,l,j}^B(x) = \frac{\big|\big\langle V_j^{(l)}, x\big\rangle + a_j^{(l)}\big|}{\big\|V_j^{(l)}\big\|_q} \quad \text{and} \quad d_{p,s}^D(x) = \frac{f_c(x) - f_s(x)}{\big\|V_c^{(L+1)} - V_s^{(L+1)}\big\|_q}, \tag{1}$$

for every $l = 1, \ldots, L$, $j = 1, \ldots, n_L$, $s = 1, \ldots, K$ and $s \neq c$, which represent the $N$ $l_p$-distances of $x$ to the hyperplanes defining the polytope $Q(x)$ and the $K - 1$ $l_p$-distances of $x$ to the hyperplanes defining the decision boundaries in $Q(x)$. Finally, we define

$$d_p^B(x) = \min_{l=1,\ldots,L}\min_{j=1,\ldots,n_l} d_{p,l,j}^B(x) \quad \text{and} \quad d_p^D(x) = \min_{\substack{s=1,\ldots,K \\ s \neq c}} d_{p,s}^D \tag{2}$$

as the minimum values of these two sets of distances (note that $d_p^D(x) < 0$ if $x$ is misclassified).

## 2.2 Robustness guarantees inside linear regions

The $l_p$-*robustness* $\mathbf{r}_p(x)$ of a classifier $f$ at a point $x$, belonging to class $c$, wrt the $l_p$-norm is defined as the optimal value of the following optimization problem

$$\mathbf{r}_p(x) = \min_{\delta \in \mathbb{R}^d} \|\delta\|_p, \quad \text{s.th.} \quad \max_{l \neq c} f_l(x + \delta) \geq f_c(x + \delta), \quad x + \delta \in S, \tag{3}$$

where is $S$ a set of constraints on the input, e.g. pixel values of images have to be in $[0, 1]$. The $l_p$-robustness $\mathbf{r}_p(x)$ is the smallest $l_p$-distance to $x$ of a point which is classified differently from $c$. Thus, $\mathbf{r}_p(x) = 0$ for misclassified points. The following theorem from Croce et al. (2019a), rephrased to fit the current notation, provides guarantees on $\mathbf{r}_p(x)$.

**Theorem 2.1 (Croce et al. (2019a))** *If* $d_p^B(x) < d_p^D(x)$, *then* $\mathbf{r}_p(x) \geq d_p^B(x)$, *while if* $|d_p^D(x)| \leq d_p^B(x)$, *then* $\mathbf{r}_p(x) = \max\{d_p^D(x), 0\}$.

Although Theorem 2.1 holds for any $l_p$-norm with $p \geq 1$, it requires to compute $d_p^B(x)$ and $d_p^D(x)$ for every $p$ individually. In this paper, exploiting this result and the geometrical arguments presented in Section 3, we show that it is possible to derive bounds on the robustness $\mathbf{r}_p(x)$ for any $p \in (1, \infty)$ using only information on $\mathbf{r}_1(x)$ and $\mathbf{r}_\infty(x)$.

In the next section, we show that the straightforward usage of standard $l_p$-norms inequalities does not yield meaningful bounds on the $l_p$-robustness inside the union of the $l_1$- and $l_\infty$-ball, since these bounds depend on the dimension of the input space of the network.

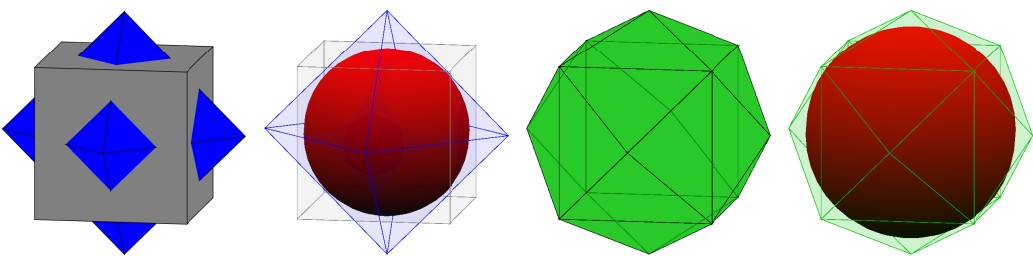

Figure 1: Visualization of the $l_2$-ball contained in the union resp. the convex hull of the union of $l_1$- and $l_\infty$-balls in $\mathbb{R}^3$. **First column**: co-centric $l_1$-ball (blue) and $l_\infty$-ball (black). **Second**: in red the largest $l_2$-ball completely contained in the union of $l_1$- and $l_\infty$-ball. **Third**: in green the convex hull of the union of the $l_1$- and $l_\infty$-ball. **Fourth**: the largest $l_2$-ball (red) contained in the convex hull. The $l_2$-ball contained in the convex hull is significantly larger than that contained in the union of $l_1$- and $l_\infty$-ball.

## 3  Minimal $l_p$-norm of the complement of the union of $l_1$- and $l_\infty$-ball and its convex hull

Let $B_1 = \{x \in \mathbb{R}^d : \|x\|_1 \le \epsilon_1\}$ and $B_\infty = \{x \in \mathbb{R}^d : \|x\|_\infty \le \epsilon_\infty\}$ be the $l_1$-ball of radius $\epsilon_1 > 0$ and the $l_\infty$-ball of radius $\epsilon_\infty > 0$ respectively, both centered at the origin in $\mathbb{R}^d$. We also assume $\epsilon_1 \in (\epsilon_\infty, d\epsilon_\infty)$, so that $B_1 \not\subseteq B_\infty$ and $B_\infty \not\subseteq B_1$.
Suppose we can guarantee that the classifier does not change its label in $U_{1,\infty} = B_1 \cup B_\infty$. Which guarantee does that imply for all intermediate $l_p$-norms? This question can be simply answered by computing the minimal $l_p$-norms over $\mathbb{R}^d \setminus U_{1,\infty}$, namely $\min_{x \in \mathbb{R}^d \setminus U_{1,\infty}} \|x\|_p$.
By the standard norm inequalities it holds, for every $x \in \mathbb{R}^d$, that

$$\|x\|_p \ge \|x\|_\infty \quad \text{and} \quad \|x\|_p \ge \|x\|_1 \, d^{\frac{1-p}{p}},$$

and thus a naive application of these inequalities yields the bound

$$\min_{x \in \mathbb{R}^d \setminus U_{1,\infty}} \|x\|_p \ge \max\left\{\epsilon_\infty, \epsilon_1 d^{\frac{1-p}{p}}\right\}. \tag{4}$$

However, this naive bound does not take into account that we know that $\|x\|_1 \ge \epsilon_1$ *and* $\|x\|_\infty \ge \epsilon_\infty$. Our first result yields the exact value taking advantage of this information.

**Proposition 3.1** *If $d \ge 2$ and $\epsilon_1 \in (\epsilon_\infty, d\epsilon_\infty)$, then*

$$\min_{x \in \mathbb{R}^d \setminus U_{1,\infty}} \|x\|_p = \left(\epsilon_\infty^p + \frac{(\epsilon_1 - \epsilon_\infty)^p}{(d-1)^{p-1}}\right)^{\frac{1}{p}}. \tag{5}$$

Thus a guarantee both for $l_1$- and $l_\infty$-ball yields a guarantee for all intermediate $l_p$-norms. However, for affine classifiers a guarantee for $B_1$ *and* $B_\infty$ implies a guarantee wrt the convex hull $C$ of their union $B_1 \cup B_\infty$. This can be seen by the fact that an affine classifier generates two half-spaces, and the convex hull of a set $A$ is the intersection of all half-spaces containing $A$. Thus, inside $C$ the decision of the affine classifier cannot change if it is guaranteed not to change in $B_1$ *and* $B_\infty$, as $C$ is completely contained in one of the half-spaces generated by the classifier (see Figure 1 for illustrations of $B_1$, $B_\infty$, their union and their convex hull). With the following theorem, we characterize, for any $p \ge 1$, the minimal $l_p$-norm over $\mathbb{R}^d \setminus C$.

**Theorem 3.1** *Let $C$ be the convex hull of $B_1 \cup B_\infty$. If $d \ge 2$ and $\epsilon_1 \in (\epsilon_\infty, d\epsilon_\infty)$, then*

$$\min_{x \in \mathbb{R}^d \setminus C} \|x\|_p = \frac{\epsilon_1}{\left(\epsilon_1/\epsilon_\infty - \alpha + \alpha^q\right)^{1/q}}, \tag{6}$$

*where $\alpha = \frac{\epsilon_1}{\epsilon_\infty} - \lfloor \frac{\epsilon_1}{\epsilon_\infty} \rfloor$ and $\frac{1}{p} + \frac{1}{q} = 1$.*

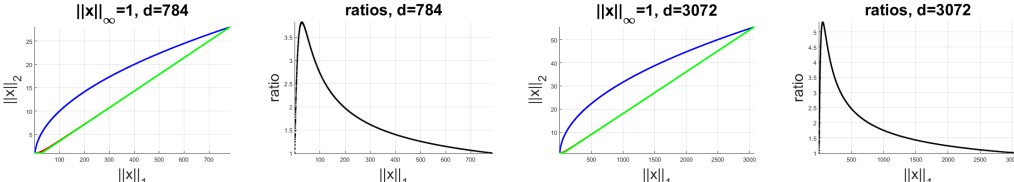

Figure 2: Comparison of the minimal $l_2$-norm over $\mathbb{R}^d \setminus C$ (6) (blue), $\mathbb{R}^d \setminus U_{1,\infty}$ (5) (red) and its naive lower bound (4) (green). We fix $\epsilon_\infty = 1$ and show the results varying $\epsilon_1 \in (1, d)$, for $d = 784$ and $d = 3072$. We plot the value (or a lower bound in case of (4)) of the minimal $\|x\|_2$, depending on $\epsilon_1$, given by the different approaches (first and third plots). The red curves are almost completely hidden by the green ones, as they mostly overlap, but can be seen for small values of $\|x\|_1$. Moreover, we report (second and fourth plots) the ratios of the minimal $\|x\|_2$ for $\mathbb{R}^d \setminus \text{conv}(B_1 \cup B_\infty)$ and $\mathbb{R}^d \setminus (B_1 \cup B_\infty)$. The values provided by (6) are much larger than those of (5).

Note that our expression in Theorem 3.1 is exact and not just a lower bound. Moreover, the minimal $l_p$-distance of $\mathbb{R}^d \setminus C$ to the origin in Equation (6) is independent from the dimension $d$, in contrast to the expression for the minimal $l_p$-norm over $\mathbb{R}^d \setminus U_{1,\infty}$ in (5) and its naive lower bound in (4), which are both decreasing for increasing $d$ and $p > 1$. In Figure 1 we compare visually the largest $l_2$-balls (in red) fitting inside either $U_{1,\infty}$ or the convex hull $C$ in $\mathbb{R}^3$, showing that the one in $C$ is clearly larger. In Figure 2 we provide a quantitative comparison in high dimensions. We plot the minimal $l_2$-norm over $\mathbb{R}^d \setminus C$ (6) (blue) and over $\mathbb{R}^d \setminus U_{1,\infty}$ (5) (red) and its naive lower bound (4) (green). We fix $\|x\|_\infty = \epsilon_\infty = 1$ and vary $\epsilon_1 \in [1, d]$, with either $d = 784$ (left) or $d = 3072$ (right), i.e. the dimensions of the input spaces of MNIST and CIFAR-10. One sees clearly that the blue line corresponding to (6) is significantly higher than the other two. In the second and fourth plots of Figure 2 we show, for each $\epsilon_1$, the ratio of the $l_2$-distances given by (6) and (5). The maximal ratio is about 3.8 for $d = 784$ and 5.3 for $d = 3072$, meaning that the advantage of (6) increases with $d$ (for a more detailed analysis see A.3).

These two examples indicate that the $l_p$-balls contained in $C$ can be a few times larger than those in $U_{1,\infty}$. Recall that we deal with piecewise affine networks. If we could enlarge the *linear regions* on which the classifier is affine so that it contains the $l_1$- and $l_\infty$-ball of some desired radii, we would automatically get the $l_p$-balls of radii given by Theorem 3.1 to fit in the linear regions. The next section formalizes the resulting robustness guarantees.

## 4    Universal provable robustness with respect to all $l_p$-norms

Combining the results of Theorems 2.1 and 3.1, in the next theorem we derive lower bounds on the robustness of a continuous piecewise affine classifier $f$, e.g. a ReLU network, at a point $x$ wrt any $l_p$-norm with $p \geq 1$ using only $d_1^B(x)$, $d_1^D(x)$, $d_\infty^B(x)$ and $d_\infty^D(x)$ (see (2)).

**Theorem 4.1** *Let $d_p^B(x)$, $d_p^D(x)$ be defined as in (2) and define $\rho_1 = \min\{d_1^B(x), |d_1^D(x)|\}$ and $\rho_\infty = \min\{d_\infty^B(x), |d_\infty^D(x)|\}$. If $d \geq 2$ and $x$ is correctly classified, then*

$$\mathbf{r}_p(x) \geq \frac{\rho_1}{\left(\rho_1/\rho_\infty - \alpha + \alpha^q\right)^{1/q}}, \tag{7}$$

*for any $p \in (1, \infty)$, with $\alpha = \frac{\rho_1}{\rho_\infty} - \lfloor \frac{\rho_1}{\rho_\infty} \rfloor$ and $\frac{1}{p} + \frac{1}{q} = 1$.*

Croce et al. (2019a) add a regularization term to the training objective in order to enlarge the values of $d_p^B(x)$ and $d_p^D(x)$ for a fixed $p$, with $x$ being the training points (note that they optimize $d_p^D(x)$ and not $|d_p^D(x)|$ to encourage correct classification).
Sorting in increasing order $d_{p,l,j}^B$ and $d_{p,s}^D$, (see (1)), that is the $l_p$-distances to the hyperplanes defining $Q(x)$ and to decision hyperplanes, and denoting them as $d_{p,\pi_i^B}^B$ and $d_{p,\pi_i^D}^D$ respectively,

the *Maximum Margin Regularizer* (MMR) of Croce et al. (2019a) is defined as

$$\text{MMR-}l_p(x) = \frac{1}{k_B} \sum_{i=1}^{k_B} \max\left(0, 1 - \frac{d_{p,\pi_i^B}^B(x)}{\gamma_B}\right) + \frac{1}{k_D} \sum_{i=1}^{k_D} \max\left(0, 1 - \frac{d_{p,\pi_i^D}^D(x)}{\gamma_D}\right). \quad (8)$$

It tries to push the $k_B$ closest hyperplanes defining $Q(x)$ farther than $\gamma_B$ from $x$ and the $k_D$ closest decision hyperplanes farther than $\gamma_D$ from $x$ both wrt the $l_p$-metric. In other words, MMR-$l_p$ aims at widening the linear regions around the training points so that they contain $l_p$-balls of radius either $\gamma_B$ or $\gamma_D$ centered in the training points. Using MMR-$l_p$ wrt a fixed $l_p$-norm, possibly in combination with the adversarial training of Madry et al. (2018), leads to classifiers which are empirically resistant wrt $l_p$-adversarial attacks and are easily verifiable by state-of-the-art methods to provide lower bounds on the true robustness.

For our goal of simultaneous $l_p$-robustness guarantees for all $p \geq 1$, we use the insights obtained from Theorem 4.1 to propose a combination of MMR-$l_1$ and MMR-$l_\infty$, called MMR-Universal. It enhances implicitly robustness wrt every $l_p$-norm without actually computing and modifying separately all the distances $d_p^B(x)$ and $d_p^D(x)$ for the different values of $p$.

**Definition 4.1 (MMR-Universal)** *Let $x$ be a training point. We define the regularizer*

$$MMR\text{-}Universal(x) = \frac{1}{k_B} \sum_{i=1}^{k_B} \lambda_1 \max\left(0, 1 - \frac{d_{1,\pi_{1,i}^B}^B(x)}{\gamma_1}\right) + \lambda_\infty \max\left(0, 1 - \frac{d_{\infty,\pi_{\infty,i}^B}^B(x)}{\gamma_\infty}\right)$$

$$+ \frac{1}{K-1} \sum_{i=1}^{K-1} \lambda_1 \max\left(0, 1 - \frac{d_{1,\pi_{1,i}^D}^D(x)}{\gamma_1}\right) + \lambda_\infty \max\left(0, 1 - \frac{d_{\infty,\pi_{\infty,i}^D}^D(x)}{\gamma_\infty}\right),$$

$$(9)$$

*where $k_B \in \{1, \ldots, N\}$, $\lambda_1, \lambda_\infty, \gamma_1, \gamma_\infty > 0$.*

We stress that, even if the formulation of MMR-Universal is based on MMR-$l_p$, it is just thanks to the novel geometrical motivation provided by Theorem 3.1 and its interpretation in terms of robustness guarantees of Theorem 4.1 that we have a theoretical justification of MMR-Universal. Moreover, we are not aware of any other approach which can enforce simultaneously $l_1$- and $l_\infty$-guarantees, which is the key property of MMR-Universal.

The loss function which is minimized while training the classifier $f$ is then, with $\{(x_i, y_i)\}_{i=1}^T$ being the training set and CE the cross-entropy loss,

$$L\left(\{(x_i, y_i)\}_{i=1}^T\right) = \frac{1}{T} \sum_{i=1}^T \text{CE}(f(x_i), y_i) + \text{MMR-Universal}(x_i).$$

During the optimization our regularizer aims at pushing both the polytope boundaries and the decision hyperplanes farther than $\gamma_1$ in $l_1$-distance and farther than $\gamma_\infty$ in $l_\infty$-distance from the training point $x$, in order to achieve robustness close or better than $\gamma_1$ and $\gamma_\infty$ respectively. According to Theorem 4.1, this enhances also the $l_p$-robustness for $p \in (1, \infty)$. Note that if the projection of $x$ on a decision hyperplane does not lie inside $Q(x)$, $d_p^D(x)$ is just an approximation of the signed distance to the true decision surface, in which case Croce et al. (2019a) argue that it is an approximation of the local Cross-Lipschitz constant which is also associated to robustness (see Hein & Andriushchenko (2017)). The regularization parameters $\lambda_1$ and $\lambda_\infty$ are used to balance the weight of the $l_1$- and $l_\infty$-term in the regularizer, and also wrt the cross-entropy loss. Note that the terms of MMR-Universal involving the quantities $d_{p,\pi_{p,i}^D}^D(x)$ penalize misclassification, as they take negative values in this case.

Moreover, we take into account the $k_B$ closest hyperplanes and not just the closest one as done in Theorems 2.1 and 4.1. This has two reasons: first, in this way the regularizer enlarges the size of the linear regions around the training points more quickly and effectively, given the large number of hyperplanes defining each polytope. Second, pushing many hyperplanes influences also the neighboring linear regions of $Q(x)$. This comes into play when, in order to get better bounds on the robustness at $x$, one wants to explore also a portion of the input space outside of the linear region $Q(x)$, which is where Theorem 4.1 holds. As noted in

Raghunathan et al. (2018); Croce et al. (2019a); Xiao et al. (2019), established methods to compute lower bounds on the robustness are loose or completely fail when using normally trained models. In fact, their effectiveness is mostly related to how many ReLU units have stable sign when perturbing the input $x$ within a given $l_p$-ball. This is almost equivalent to having the hyperplanes far from $x$ in $l_p$-distance, which is what MMR-Universal tries to accomplish. This explains why in Section 5 we can certify the models trained with MMR-Universal with the methods of Wong & Kolter (2018) and Tjeng et al. (2019).

## 5 EXPERIMENTS

We compare the models obtained via our MMR-Universal regularizer[1] to state-of-the-art methods for provable robustness and adversarial training. As evaluation criterion we use the *robust test error*, defined as the largest classification error when every image of the test set can be perturbed within a fixed set (e.g. an $l_p$-ball of radius $\epsilon_p$). We focus on the $l_p$-balls with $p \in \{1, 2, \infty\}$. Since computing the robust test error is in general an NP-hard problem, we evaluate lower and upper bounds on it. The lower bound is the fraction of points for which an attack can change the decision with perturbations in the $l_p$-balls of radius $\epsilon_p$ (adversarial samples), that is with $l_p$-norm smaller than $\epsilon_p$. For this task we use the PGD-attack (Kurakin et al. (2017); Madry et al. (2018); Tramèr & Boneh (2019)) and the FAB-attack (Croce & Hein (2019)) for $l_1$, $l_2$ and $l_\infty$, MIP (Tjeng et al. (2019)) for $l_\infty$ and the Linear Region Attack (Croce et al. (2019b)) for $l_2$ and apply all of them (see C.3 for details). The upper bound is the portion of test points for which we cannot certify, using the methods of Tjeng et al. (2019) and Wong & Kolter (2018), that no $l_p$-perturbation smaller than $\epsilon_p$ can change the correct class of the original input.

Smaller values of the upper bounds on the robust test error indicate models with better provable robustness. While lower bounds give an empirical estimate of the true robustness, it has been shown that they can heavily underestimate the vulnerability of classifiers (e.g. by Athalye et al. (2018); Mosbach et al. (2018)).

### 5.1 CHOICE OF $\epsilon_p$

In choosing the values of $\epsilon_p$ for $p \in \{1, 2, \infty\}$, we try to be consistent with previous literature (e.g. Wong & Kolter (2018); Croce et al. (2019a)) for the values of $\epsilon_\infty$ and $\epsilon_2$. Equation (6) provides, given $\epsilon_1$ and $\epsilon_\infty$, a value at which one can expect $l_2$-robustness (approximately $\epsilon_2 = \sqrt{\epsilon_1 \epsilon_\infty}$). Then we fix $\epsilon_1$ such that this approximation is slightly larger than the desired $\epsilon_2$. We show in Table 1 the values chosen for $\epsilon_p$, $p \in \{1, 2, \infty\}$, and used to compute the robust test error in Table 2. Notice that for these values no $l_p$-ball is contained in the others.

Table 1: The values chosen for $\epsilon_p$ on the different datasets and the expected $l_2$-robustness level (last column) given $\epsilon_1$ and $\epsilon_\infty$, computed according to (6).

| dataset | $\epsilon_1$ | $\epsilon_\infty$ | $\epsilon_2$ | $\epsilon_2$ by (6) |
|---|---|---|---|---|
| MNIST / F-MNIST | 1 | 0.1 | 0.3 | 0.3162 |
| GTS | 3 | $4/255$ | 0.2 | 0.2170 |
| CIFAR-10 | 2 | $2/255$ | 0.1 | 0.1252 |

Moreover, we compute for the plain models the percentage of adversarial examples given by an $l_1$-attack (we use the PGD-attack) with budget $\epsilon_1$ which have also $l_\infty$-norm smaller than or equal to $\epsilon_\infty$, and vice versa. These percentages are zero for all the datasets, meaning that being (provably) robust in the union of these $l_p$-balls is much more difficult than in just one of them (see also C.1).

Table 2: We report, for the different datasets and training schemes, the test error (TE) and lower (LB) and upper (UB) bounds on the robust test error (in percentage) wrt the union of $l_p$-norms for $p \in \{1, 2, \infty\}$ denoted as $l_1 + l_2 + l_\infty$ (that is the largest test error possible if any perturbation in the union $l_1 + l_2 + l_\infty$ is allowed). The training schemes compared are plain training, adversarial trainings of Madry et al. (2018); Tramèr & Boneh (2019) (AT), robust training of Wong & Kolter (2018); Wong et al. (2018) (KW), MMR regularization of Croce et al. (2019a), MMR combined with AT (MMR+AT) and our MMR-Universal regularization. The models of our MMR-Universal are the only ones which have non trivial upper bounds on the robust test error for all datasets.

**provable robustness against multiple perturbations**

| model | | TE | $l_1 + l_2 + l_\infty$ LB | UB | | TE | $l_1 + l_2 + l_\infty$ LB | UB |
|---|---|---|---|---|---|---|---|---|
| plain | | 0.85 | 88.5 | 100 | | 9.32 | 100 | 100 |
| AT-$l_\infty$ | | 0.82 | 4.7 | 100 | | 11.54 | 26.3 | 100 |
| AT-$l_2$ | | 0.87 | 25.9 | 100 | | 8.10 | 98.8 | 100 |
| AT-$(l_1, l_2, l_\infty)$ | | 0.80 | 4.9 | 100 | | 14.13 | 29.6 | 100 |
| KW-$l_\infty$ | MNIST | 1.21 | 4.8 | 100 | F-MNIST | 21.73 | 43.6 | 100 |
| KW-$l_2$ | | 1.11 | 10.3 | 100 | | 13.08 | 66.7 | 86.8 |
| MMR-$l_\infty$ | | 1.65 | 10.4 | 100 | | 14.51 | 36.7 | 100 |
| MMR-$l_2$ | | 2.57 | 78.6 | 99.9 | | 12.85 | 95.8 | 100 |
| MMR+AT-$l_\infty$ | | 1.19 | 4.1 | 100 | | 14.52 | 31.8 | 100 |
| MMR+AT-$l_2$ | | 1.73 | 15.3 | 99.9 | | 13.40 | 66.5 | 99.1 |
| MMR-Universal | | 3.04 | 12.4 | **20.8** | | 18.57 | 43.5 | **52.9** |
| plain | | 6.77 | 71.5 | 100 | | 23.29 | 88.6 | 100 |
| AT-$l_\infty$ | | 6.83 | 64.0 | 100 | | 27.06 | 52.5 | 100 |
| AT-$l_2$ | | 8.76 | 59.0 | 100 | | 25.84 | 62.1 | 100 |
| AT-$(l_1, l_2, l_\infty)$ | | 8.80 | 45.2 | 100 | | 35.41 | 57.1 | 100 |
| KW-$l_\infty$ | GTS | 15.57 | 87.8 | 100 | CIFAR-10 | 38.91 | 51.9 | 100 |
| KW-$l_2$ | | 14.35 | 57.6 | 100 | | 40.24 | 54.0 | 100 |
| MMR-$l_\infty$ | | 13.32 | 71.3 | 99.6 | | 34.61 | 58.7 | 100 |
| MMR-$l_2$ | | 14.21 | 62.6 | 80.9 | | 40.93 | 72.9 | 98.0 |
| MMR+AT-$l_\infty$ | | 14.89 | 82.8 | 100 | | 35.38 | 50.8 | 100 |
| MMR+AT-$l_2$ | | 15.34 | 58.1 | 84.8 | | 37.78 | 61.3 | 99.9 |
| MMR-Universal | | 15.98 | 51.6 | **52.4** | | 46.96 | 63.8 | **64.6** |

## 5.2 MAIN RESULTS

We train CNNs on MNIST, Fashion-MNIST (Xiao et al. (2017)), German Traffic Sign (GTS) (Stallkamp et al. (2012)) and CIFAR-10 (Krizhevsky et al. (2014)). We consider several training schemes: plain training, the PGD-based adversarial training (AT) of Madry et al. (2018) and its extension to multiple $l_p$-balls in Tramèr & Boneh (2019), the robust training (KW) of Wong & Kolter (2018); Wong et al. (2018), the MMR-regularized training (MMR) of Croce et al. (2019a), either alone or with adversarial training (MMR+AT) and the training with our regularizer MMR-Universal. We use AT, KW, MMR and MMR+AT wrt $l_2$ and $l_\infty$, as these are the norms for which such methods have been used in the original papers. More details about the architecture and models in C.3.

In Table 2 we report test error (TE) computed on the whole test set and lower (LB) and upper (UB) bounds on the robust test error obtained considering the union of the three $l_p$-balls, indicated by $l_1 + l_2 + l_\infty$ (these statistics are on the first 1000 points of the test set). The lower bounds $l_1 + l_2 + l_\infty$-LB are given by the fraction of test points for which one of the adversarial attacks wrt $l_1$, $l_2$ and $l_\infty$ is successful. The upper bounds $l_1 + l_2 + l_\infty$-UB are computed as the percentage of points for which at least one of the three $l_p$-balls is not certified to be free of adversarial examples (lower is better). This last one is the metric of main interest, since we aim at *universally provably robust* models. In C.2 we report the lower and upper bounds for the individual norms for every model.

---

[1]Code available at `https://github.com/fra31/mmr-universal`.

MMR-Universal is the only method which can give non-trivial upper bounds on the robust test error for all datasets, while almost all other methods aiming at provable robustness have $l_1 + l_2 + l_\infty$-UB close to or at 100%. Notably, on GTS the upper bound on the robust test error of MMR-Universal is lower than the lower bound of all other methods except AT-$(l_1, l_2, l_\infty)$, showing that MMR-Universal *provably* outperforms existing methods which provide guarantees wrt individual $l_p$-balls, either $l_2$ or $l_\infty$, when certifying the union $l_1 + l_2 + l_\infty$. The test error is slightly increased wrt the other methods giving provable robustness, but the same holds true for combined adversarial training AT-$(l_1, l_2, l_\infty)$ compared to standard adversarial training AT-$l_2/l_\infty$. We conclude that MMR-Universal is the only method so far being able to provide non-trivial robustness guarantees for multiple $l_p$-balls in the case that none of them contains any other.

## 6 Conclusion

With MMR-Universal we have proposed the first method providing provable robustness guarantees for all $l_p$-balls with $p \geq 1$. Compared to existing works guaranteeing robustness wrt either $l_2$ or $l_\infty$, providing guarantees wrt the union of different $l_p$-balls turns out to be considerably harder. It is an interesting open question if the ideas developed in this paper can be integrated into other approaches towards provable robustness.

## Acknowledgements

We would like to thank Maksym Andriushchenko for helping us to set up and adapt the original code for MMR. We acknowledge support from the German Federal Ministry of Education and Research (BMBF) through the Tübingen AI Center (FKZ: 01IS18039A). This work was also supported by the DFG Cluster of Excellence "Machine Learning – New Perspectives for Science", EXC 2064/1, project number 390727645, and by DFG grant 389792660 as part of TRR 248.

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

# A    Minimal $l_p$-norm of the complement of the union of $l_1$- and $l_\infty$-ball and its convex hull

## A.1    Proof of Proposition 3.1

*Proof.* We first note that for $\epsilon_1 < \epsilon_\infty$ it holds $B_1 \subset B_\infty$ and the proof follows from the standard inequality $\|x\|_p \geq \|x\|_\infty$ where equality is attained for $x = \epsilon_\infty e_i$, where $e_i$ are standard basis vectors. Moreover, if $\epsilon_1 > d\epsilon_\infty$ it holds $B_\infty \subset B_1$ as $\max_{x \in B_\infty} \|x\|_1 = d\epsilon_\infty$ and the result follows by $\|x\|_p \geq \|x\|_1 d^{\frac{1-p}{p}}$. The equality is realized by the vector with all the entries equal to $\frac{\epsilon_1}{d}$.

For the second case we first note that using Hölder inequality $|\langle u, v \rangle| \leq \|u\|_p \|v\|_q$ where $\frac{1}{p} + \frac{1}{q} = 1$, it holds

$$\sum_{i=1}^{k} |x_i| \leq \left( \sum_{i=1}^{k} |x_i|^p \right)^{\frac{1}{p}} k^{\frac{1}{q}}.$$

Let $x \in \mathbb{R}^d$. Without loss of generality after a potential permutation of the coordinates it holds $|x_d| = \|x\|_\infty$. Then we get

$$\|x\|_p^p = \sum_{i=1}^{d} |x_i|^p = |x_d|^p + \sum_{i=1}^{d-1} |x_i|^p \geq |x_d|^p + \frac{\left( \sum_{i=1}^{d-1} |x_i| \right)^p}{(d-1)^{\frac{p}{q}}}.$$

We have

$$\min_{\|x\|_\infty \geq \epsilon_\infty, \|x\|_1 \geq \epsilon_1} |x_d|^p + \frac{\left( \sum_{i=1}^{d-1} |x_i| \right)^p}{(d-1)^{\frac{p}{q}}} = \epsilon_\infty^p + \frac{(\epsilon_1 - \epsilon_\infty)^p}{(d-1)^{p-1}},$$

noting that $|x_d| = \|x\|_\infty$ and $\sum_{i=1}^{d-1} |x_i| \geq \epsilon_1 - \epsilon_\infty$.
Finally, we note that the vector

$$v = \sum_{i=1}^{d-1} \frac{\epsilon_1 - \epsilon_\infty}{d-1} e_i + \epsilon_\infty e_d,$$

realizes equality. Indeed, $\|v\|_p^p = (d-1) \frac{(\epsilon_1 - \epsilon_\infty)^p}{(d-1)^p} + \epsilon_\infty^p$, which finishes the proof. $\qquad \square$

## A.2    Proof of Theorem 3.1

*Proof.* We first note that the minimum of the $l_p$-norm over $\mathbb{R}^d \setminus C$ lies on the boundary of $C$ (otherwise any point on the segment joining the origin and $y$ and outside $C$ would have $l_p$-norm smaller than $y$). Moreover, the faces of $C$ are contained in hyperplanes constructed as the affine hull of a subset of $d$ points from the union of the vertices of $B_1$ and $B_\infty$.
The vertices of $B_1$ are $V_1 = \{\epsilon_1 e_i, -\epsilon_1 e_i \,|\, i = 1, \dots, d\}$, where $e_i$ is the $i$-th element of the standard basis of $\mathbb{R}^d$, and that of $B_\infty$ are $V_\infty$, consisting of the $2^d$ vectors whose components are elements of $\{\epsilon_\infty, -\epsilon_\infty\}$. Note that $V_1 \cap V_\infty = \emptyset$. Any subset of $d$ vertices from $V_1 \cup V_\infty$ defines a hyperplane which contains a face of $C$ if it does not contain any point of the interior of $C$.

Let $S$ be a set of vertices defining a hyperplane containing a face of $C$. We first derive conditions on the vertices contained in $S$. Let $k = \left\lfloor \frac{\epsilon_1}{\epsilon_\infty} \right\rfloor \in \mathbb{N}$ and $\alpha = \frac{\epsilon_1}{\epsilon_\infty} - k \in [0, 1)$. Note that $k + 1 > \frac{\epsilon_1}{\epsilon_\infty}$. Then no more than $k$ vertices of $B_1$ belong to $S$, that is to a face of $C$. In fact, if we consider $k + 1$ vertices of $B_1$, namely wlog $\{\epsilon_1 e_1, \dots, \epsilon_1 e_{k+1}\}$, and consider their convex combination $z = \epsilon_1 \sum_{i=1}^{k+1} \frac{1}{k+1} e_i$ then $\|z\|_\infty = \frac{\epsilon_1}{k+1} < \epsilon_\infty$ by the definition of $k$. Thus $S$ cannot contain more than $k$ vertices of $B_1$.

Second, assume $\epsilon_1 e_j$ is in $S$. If any vertex $v$ of $B_\infty$ with $v_j = -\epsilon_\infty$ is also in $S$ then, with $\alpha' = \frac{\epsilon_\infty}{\epsilon_1 + \epsilon_\infty} \in (0, 1)$, we get

$$\|\alpha' \epsilon_1 e_j + (1 - \alpha') v\|_\infty = \max\{|\alpha' \epsilon_1 - (1 - \alpha') \epsilon_\infty|, (1 - \alpha') \epsilon_\infty\},$$

where $(1 - \alpha')\epsilon_\infty < \epsilon_\infty$ and

$$|\alpha'\epsilon_1 - (1 - \alpha')\epsilon_\infty| = |\alpha'(\epsilon_1 + \epsilon_\infty) - \epsilon_\infty| = 0 < \epsilon_\infty.$$

Thus $S$ would not span a face as a convex combination intersects the interior of $C$. This implies that if $\epsilon_1 e_j$ is in S then all the vertices $v$ of $B_\infty$ in $S$ need to have $v_j = \epsilon_\infty$, otherwise $S$ would not define a face of $C$. Analogously, if $-\epsilon_1 e_j \in S$ then any vertex $v$ of $B_\infty$ in $S$ has $v_j = -\epsilon_\infty$. However, we note that out of symmetry reasons we can just consider faces of $C$ in the positive orthant and thus we consider in the following just sets $S$ which contain vertices of "positive type" $\epsilon_1 e_j$.

Let now $S$ be a set (not necessarily defining a face of $C$) containing $h \leq k$ vertices of $B_1$ and $d - h$ vertices of $B_\infty$ and $P$ the matrix whose columns are these points. The matrix $P$ has the form

$$P = \left( \begin{array}{cccc|ccc}
\epsilon_1 & 0 & \ldots & 0 & \epsilon_\infty & \ldots & \epsilon_\infty \\
0 & \epsilon_1 & \ldots & 0 & \epsilon_\infty & \ldots & \epsilon_\infty \\
\ldots & & & & & & \\
0 & \ldots & 0 & \epsilon_1 & \epsilon_\infty & \ldots & \epsilon_\infty \\
\hline
0 & \ldots & & 0 & & & \\
\ldots & & & & & A & \\
0 & \ldots & & 0 & & &
\end{array} \right)$$

where $A \in \mathbb{R}^{d-h,d-h}$ is a matrix whose entries are either $\epsilon_\infty$ or $-\epsilon_\infty$. If the matrix $P$ does not have full rank then the origin belongs to any hyperplane containing $S$, which means it cannot be a face of $C$. This also implies $A$ has full rank if $S$ spans a face of $C$.

We denote by $\pi$ the hyperplane generated by the affine hull of $S$ (the columns of $P$) assuming that $A$ has full rank. Every point $b$ belonging to the hyperplane $\pi$ generated by $S$ is such that there exists a unique $a \in \mathbb{R}^d$ which satisfies

$$P'a = \left( \begin{array}{c} \mathbf{1}_{1,d} \\ P \end{array} \right) a = \left( \begin{array}{cccc|ccc}
1 & & \ldots & & & & 1 \\
\epsilon_1 & 0 & \ldots & 0 & \epsilon_\infty & \ldots & \epsilon_\infty \\
0 & \epsilon_1 & \ldots & 0 & \epsilon_\infty & \ldots & \epsilon_\infty \\
\ldots & & & & & & \\
0 & \ldots & 0 & \epsilon_1 & \epsilon_\infty & \ldots & \epsilon_\infty \\
\hline
0 & \ldots & & 0 & & & \\
\ldots & & & & & A & \\
0 & \ldots & & 0 & & &
\end{array} \right) a = \left( \begin{array}{c} 1 \\ b \end{array} \right) = b',$$

where $\mathbf{1}_{d_1,d_2}$ is the matrix of size $d_1 \times d_2$ whose entries are 1.

The matrix $(P', b') \in \mathbb{R}^{d+1,d+1}$ need not have full rank, so that

$$\mathrm{rank} P' = \mathrm{rank}(P', b') = \dim a = d$$

and then the linear system $P'a = b'$ has a unique solution.
We define the vector $v \in \mathbb{R}^d$ as solution of $P^T v = \mathbf{1}_{d,1}$, which is unique as $P$ has full rank. From their definitions we have $Pa = b$ and $\mathbf{1}^T a = 1$, so that

$$1 = \mathbf{1}^T a = (P^T v)^T a = v^T P a = v^T b,$$

and thus

$$\langle b, v \rangle = 1, \tag{10}$$

noticing that this also implies that any vector $b \in \mathbb{R}^d$ such that $\langle b, v \rangle = 1$ belongs to $\pi$ (suppose that $\exists q \notin \pi$ with $\langle q, v \rangle = 1$, then define $c$ as the solution of $Pc = q$ and then $1 = \langle q, v \rangle = \langle Pc, v \rangle = \langle c, P^T v \rangle = \langle c, \mathbf{1} \rangle$ which contradicts that $q \notin \pi$).
Applying Hölder inequality to (10) we get for any $b \in \pi$,

$$\|b\|_p \geq \frac{1}{\|v\|_q}, \tag{11}$$

where $\frac{1}{p} + \frac{1}{q} = 1$. Moreover, as $p \in (1, \infty)$ there exists always a point $b^*$ for which (11) holds as equality.

In the rest of the proof we compute $\|v\|_q$ for any $q > 1$ when $S$ is a face of $C$ and then (11) yields the desired minimal value of $\|b\|_p$ over all $b$ lying in faces of $C$.

Let $v = (v_1, v_2)$, $v_1 \in \mathbb{R}^h$, $v_2 \in \mathbb{R}^{d-h}$ and $I_h$ denotes the identity matrix of $\mathbb{R}^{h,h}$. It holds

$$P^T v = \begin{pmatrix} \epsilon_1 I_h & 0 \\ \epsilon_\infty \mathbf{1}_{d-h,h} & A^T \end{pmatrix} \begin{pmatrix} v_1 \\ v_2 \end{pmatrix} = \begin{pmatrix} \mathbf{1}_{h,1} \\ \mathbf{1}_{d-h,1} \end{pmatrix} = \mathbf{1}_{d,1},$$

which implies

$$v_1 = \left( \frac{1}{\epsilon_1}, \dots, \frac{1}{\epsilon_1} \right) \quad \text{and} \quad A^T v_2 = \mathbf{1}_{d-h,1} - \frac{h\epsilon_\infty}{\epsilon_1} \mathbf{1}_{d-h,1} = \left( 1 - h\frac{\epsilon_\infty}{\epsilon_1} \right) \mathbf{1}_{d-h,1}.$$

Moreover, we have

$$\|v\|_1 = \|v_1\|_1 + \|v_2\|_1 = \frac{h}{\epsilon_1} + \|v_2\|_1, \quad \|v\|_\infty = \max \left\{ \frac{1}{\epsilon_1}, \|v_2\|_\infty \right\}. \tag{12}$$

If $S$ generates a face, then by definition $\pi$ does not intersect with the interior of $C$ and thus it holds for all $b \in \pi$: $\|b\|_1 \geq \epsilon_1$ and $\|b\|_\infty \geq \epsilon_\infty$. Suppose $\|v\|_1 = c > \frac{1}{\epsilon_\infty}$. Then there exists $b^* \in \pi$ such equality in Hölder's equality is realized, that is $1 = \langle b^*, v \rangle = \|b^*\|_\infty \|v\|_1$, and thus $\|b^*\|_\infty = \frac{1}{c} < \epsilon_\infty$, which contradicts $\|b\|_\infty \geq \epsilon_\infty$ for all $b \in \pi$ and thus it must hold $\|v\|_1 \leq \frac{1}{\epsilon_\infty}$. Similarly, one can derive $\|v\|_\infty \leq \frac{1}{\epsilon_1}$. Combining (12) with the just derived inequalities we get upper bounds on the norms of $v_2$,

$$\|v_2\|_1 \leq \frac{1}{\epsilon_\infty} - \frac{h}{\epsilon_1} \quad \text{and} \quad \|v_2\|_\infty \leq \frac{1}{\epsilon_1}. \tag{13}$$

Furthermore $v_2$ is defined as the solution of

$$\frac{A^T}{\epsilon_\infty} v_2 = \left( \frac{1}{\epsilon_\infty} - \frac{h}{\epsilon_1} \right) \mathbf{1}_{d-h,1}.$$

We note that all the entries of $\frac{A^T}{\epsilon_\infty}$ are either 1 or $-1$, so that the inner product between each row of $A^T$ and $v_2$ is a lower bound on the $l_1$-norm of $v_2$. Since every entry of the r.h.s. of the linear system is $\frac{1}{\epsilon_\infty} - \frac{h}{\epsilon_1}$ we get $\|v_2\|_1 \geq \frac{1}{\epsilon_\infty} - \frac{h}{\epsilon_1}$, which combined with (13) leads to $\|v_2\|_1 = \frac{1}{\epsilon_\infty} - \frac{h}{\epsilon_1}$.

This implies that $\frac{A^T}{\epsilon_\infty} v_2 = \|v_2\|_1$. In order to achieve equality $\langle u, v \rangle = \|v\|_1$ it has to hold $u_i = \text{sgn}(v_i)$ for every $v_i \neq 0$. If at least two components of $v$ were non-zero, the corresponding columns of $A^T$ would be identical, which contradicts the fact that $A^T$ has full rank. Thus $v_2$ can only have one non-zero component which in absolute value is equal to $\frac{1}{\epsilon_\infty} - \frac{h}{\epsilon_1}$ Thus, after a potential reordering of the components, $v$ has the form

$$v = \left( \underbrace{\frac{1}{\epsilon_1}, \dots, \frac{1}{\epsilon_1}}_{h \text{ times}}, \frac{1}{\epsilon_\infty} - \frac{h}{\epsilon_1}, 0, \dots, 0 \right).$$

From the second condition in (13), we have $\frac{1}{\epsilon_\infty} - \frac{h}{\epsilon_1} \leq \frac{1}{\epsilon_1}$ and $h + 1 \geq \frac{\epsilon_1}{\epsilon_\infty} = k + \alpha$. Recalling $h \leq k$, we have

$$h \in [k + \alpha - 1, k] \cap \mathbb{N}.$$

This means that, in order for $S$ to define a face of $C$, we need $h = k$ if $\alpha > 0$, $h \in \{k-1, k\}$ if $\alpha = 0$ (in this case choosing $h = k - 1$ or $h = k$ leads to the same $v$, so in practice it is possible to use simply $h = k$ for any $\alpha$).

Once we have determined $v$, we can use again (10) and (11) to see that

$$\|b\|_p \geq \frac{1}{\|v\|_q} = \frac{1}{\left( \frac{k}{\epsilon_1^q} + \left( \frac{1}{\epsilon_\infty} - \frac{k}{\epsilon_1} \right)^q \right)^{\frac{1}{q}}} = \frac{\epsilon_1}{\left( \epsilon_1/\epsilon_\infty - \alpha + \alpha^q \right)^{1/q}}. \tag{14}$$

Finally, for any $v$ there exists $b^* \in \pi$ for which equality is achieved in (14). Suppose that this $b^*$ does not lie in a face of $C$. Then one could just consider the line segment from the origin to $b^*$ and the point intersecting the boundary of $C$ would have smaller $l_p$-norm contradicting the just derived inequality. Thus the $b^*$ realizing equality in (14) lies in a face of $C$. $\qquad\square$

### A.3 COMPARISON OF THE ROBUSTNESS GUARANTEE FOR THE UNION OF $B_1$ AND $B_\infty$ IN (5) AND THE CONVEX HULL OF $B_1$ AND $B_\infty$ IN (6)

We compare the robustness guarantees obtained by considering the union of $B_1$ and $B_\infty$ (denoted by $b_U$ in the following), see (5), and the convex hull of $B_1$ and $B_\infty$ (denoted by $b_C$ in the following), see (6). In particular, we want to determine the ratio $\delta = \frac{\epsilon_1}{\epsilon_\infty} \in [1, d]$ for which the gain in the robustness guarantee $b_C$ for the convex hull is maximized compared to just considering the robustness guarantee $b_U$ as a function of the dimension $d$ of the input space. We restrict here the analysis to the case of $p = 2$, that is computing the radius of the largest $l_2$-ball fitting inside $U_{1,\infty}$ or its convex hull $C$. Let us denote

$$b_U = \min_{x \in \mathbb{R}^d \setminus U_{1,\infty}} \|x\|_p = \left( \epsilon_\infty^p + \frac{(\epsilon_1 - \epsilon_\infty)^p}{(d-1)^{p-1}} \right)^{\frac{1}{p}}, \quad b_C = \min_{x \in \mathbb{R}^d \setminus C} \|x\|_p = \frac{\epsilon_1}{\left( \epsilon_1/\epsilon_\infty - \alpha + \alpha^q \right)^{1/q}}$$

the two bounds from (5) and (6) respectively, which can be rewritten as

$$b_U(\delta) = \epsilon_\infty \left( 1 + \frac{(\delta - 1)^p}{(d-1)^{p-1}} \right)^{\frac{1}{p}}, \quad b_C(\delta) = \frac{\epsilon_\infty \delta}{(\delta - \alpha + \alpha^q)^{1/q}} \sim \epsilon_\infty \delta^{\frac{1}{p}},$$

where $\alpha = \delta - \lfloor \delta \rfloor$. We note that

$$\delta - 1 \leq \delta - \alpha + \alpha^q = \lfloor \delta \rfloor + (\delta - \lfloor \delta \rfloor)^q \leq \delta.$$

As the differences are very small, we use instead the lower bound

$$b_C^*(\delta) = \frac{\epsilon_\infty \delta}{(\delta)^{1/q}} = \epsilon_\infty \delta^{\frac{1}{p}}.$$

We want to find the value $\delta^*$ which maximizes $\frac{b_C^*}{b_U}(\delta)$ varying $d$ (a numerical evaluation is presented in Figure 2). Notice first that $\delta^*$ maximizes also

$$\frac{(b_C^*(\delta))^p}{(b_U(\delta))^p} = \delta \left( 1 + \frac{(\delta - 1)^p}{(d-1)^{p-1}} \right)^{-1} \geq 1$$

and can be found as the solution of

$$\frac{\partial}{\partial \delta} \frac{(b_C^*(\delta))^p}{(b_U(\delta))^p} = \frac{\partial}{\partial \delta} l \left[ \delta \left( 1 + \frac{(\delta - 1)^p}{(d-1)^{p-1}} \right)^{-1} \right]$$

$$= \left( 1 + \frac{(\delta - 1)^p}{(d-1)^{p-1}} \right)^{-1} - \delta \left( 1 + \frac{(\delta - 1)^p}{(d-1)^{p-1}} \right)^{-2} \frac{p(\delta - 1)^{p-1}}{(d-1)^{p-1}} = 0,$$

which is equivalent to

$$(d-1)^{p-1} + (\delta - 1)^p - p\delta(\delta - 1)^{p-1} = 0.$$

Restricting the analysis to $p = 2$ for simplicity, we get

$$d - 1 + (\delta - 1)^2 - 2\delta(\delta - 1) = -\delta^2 + d = 0 \quad \Longrightarrow \quad \delta^* = \sqrt{d},$$

and one can check that $\delta^*$ is indeed a maximizer. Moreover, at $\delta^*$ we have a ratio between the two bounds

$$\frac{b_C}{b_U}\bigg|_{\delta^*} \geq \frac{b_C^*}{b_U}\bigg|_{\delta^*} = d^{\frac{1}{4}} \left( 1 + \frac{(\sqrt{d} - 1)^2}{(d-1)} \right)^{-\frac{1}{2}} \sim \frac{d^{\frac{1}{4}}}{\sqrt{2}}.$$

We observe that the improvement of the robustness guarantee by considering the convex hull instead of the union is increasing with dimension and is $\approx 3.8$ for $d = 784$ and $\approx 5.3$ for $d = 3072$. Thus in high dimensions there is a considerable gain by considering the convex hull.

# B UNIVERSAL PROVABLE ROBUSTNESS WITH RESPECT TO ALL $l_p$-NORMS

## B.1 PROOF OF THEOREM 4.1

*Proof.* From the definition of $d_p^B(x)$ and $d_p^D(x)$ we know that none of the hyperplanes $\{\pi_j\}_j$ (either boundaries of the polytope $Q(x)$ or decision hyperplanes) identified by $V^{(l)}$ and $v^{(l)}$, $l = 1, \ldots, L + 1$, is closer than $\min\{d_p^B(x), |d_p^D(x)|\}$ in $l_p$-distance. Therefore the interior of the $l_1$-ball of radius $\rho_1$ (namely, $B_1(x, \rho_1)$) and of the $l_\infty$-ball of radius $\rho_\infty$ ($B_\infty(x, \rho_\infty)$) centered in $x$ does not intersect with any of those hyperplanes. This implies that $\{\pi_j\}_j$ are intersecting the closure of $\mathbb{R}^d \setminus \mathrm{conv}(B_1(x, \rho_1) \cup B_\infty(x, \rho_\infty))$. Then, from Theorem 3.1 we get

$$\min\{d_p^B(x), |d_p^D(x)|\} \geq \frac{\rho_1}{\left(\rho_1/\rho_\infty - \alpha + \alpha^q\right)^{1/q}}.$$

Finally, exploiting Theorem 2.1, $\mathbf{r}_p(x) \geq \min\{d_p^B(x), |d_p^D(x)|\}$ holds. $\qquad\square$

# C EXPERIMENTS

## C.1 CHOICE OF $\epsilon_p$

In Table 3 we compute the percentage of adversarial perturbations given by the PGD-attack wrt $l_p$ with budget $\epsilon_p$ which have $l_q$-norm smaller than $\epsilon_q$, for $q \neq p$ (the values of $\epsilon_p$ and $\epsilon_q$ used are those from Table 1). We used the plain model of each dataset.

The most relevant statistics of Table 3 are about the relation between the $l_1$- and $l_\infty$-perturbations (first two rows). In fact, none of the adversarial examples wrt $l_1$ is contained in the $l_\infty$-ball, and vice versa. This means that, although the volume of the $l_1$-ball is much smaller, even because of the intersection with the box constraints $[0, 1]^d$, than that of the $l_\infty$-ball in high dimension, and most of it is actually contained in the $l_\infty$-ball, the adversarial examples found by $l_1$-attacks are anyway very different from those got by $l_\infty$-attacks. The choice of such $\epsilon_p$ is then meaningful, as the adversarial perturbations we are trying to prevent wrt the various norms are non-overlapping and in practice exploit regions of the input space significantly diverse one from another.
Moreover, one can see that also the adversarial manipulations wrt $l_1$ and $l_2$ do not overlap. Regarding the case of $l_2$ and $l_\infty$, for MNIST and F-MNIST it happens that the adversarial examples wrt $l_2$ are contained in the $l_\infty$-ball. However, as one observes in Table 4, being able to certify the $l_\infty$-ball is not sufficient to get non-trivial guarantees wrt $l_2$. In fact, all the models trained on these datasets to be provably robust wrt the $l_\infty$-norm, that is KW-$l_\infty$, MMR-$l_\infty$ and MMR+AT-$l_\infty$, have upper bounds on the robust test error in the $l_2$-ball larger than 99%, despite the values of the lower bounds are small (which means that the attacks could not find adversarial perturbations for many points).

Such analysis confirms that empirical and provable robustness are two distinct problems, and the interaction of different kinds of perturbations, as we have, changes according to which of these two scenarios one considers.

Table 3: Percentage of $l_1$-adversarial examples contained in the $l_\infty$-ball and vice versa.

|  | MNIST | F-MNIST | GTS | CIFAR-10 |
|---|---|---|---|---|
| $l_1$-perturbations with $l_\infty$-norm $\leq \epsilon_\infty$ (%) | 0.0 | 0.0 | 0.0 | 0.0 |
| $l_\infty$-perturbations with $l_1$-norm $\leq \epsilon_1$ (%) | 0.0 | 0.0 | 0.0 | 0.0 |
| $l_1$-perturbations with $l_2$-norm $\leq \epsilon_2$ (%) | 4.5 | 7.4 | 0.0 | 0.0 |
| $l_2$-perturbations with $l_1$-norm $\leq \epsilon_1$ (%) | 0.0 | 0.0 | 0.0 | 0.0 |
| $l_2$-perturbations with $l_\infty$-norm $\leq \epsilon_\infty$ (%) | 100.0 | 100.0 | 0.3 | 16.0 |
| $l_\infty$-perturbations with $l_2$-norm $\leq \epsilon_2$ (%) | 0.0 | 0.0 | 0.0 | 0.0 |

## C.2   MAIN RESULTS

In Table 4 we report, for each dataset, the test error and upper and lower bounds on the robust test error, together with the $\epsilon_p$ used, for each norm individually. It is clear that training for provable $l_p$-robustness (expressed by the upper bounds) does not, in general, yield provable $l_q$-robustness for $q \neq p$, even in the case where the lower bounds are small for both $p$ and $q$.

In order to compute the upper bounds on the robust test error in Tables 2 and 4 we use the method of Wong & Kolter (2018) for all the three $l_p$-norms and that of Tjeng et al. (2019) only for the $l_\infty$-norm. This second one exploits a reformulation of the problem in (3) in terms of mixed integer programming (MIP), which is able to exactly compute the solution of (3) for $p \in \{1, 2, \infty\}$. However, such technique is strongly limited by its high computational cost. The only reason why it is possible to use it in practice is the exploitation of some presolvers which are able to reduce the complexity of the MIP. Unfortunately, such presolvers are effective just wrt $l_\infty$. On the other hand, the method of Wong & Kolter (2018) applies directly to every $l_p$-norm. This explains why the bounds provided for $l_\infty$ are tighter than those for $l_1$ and $l_2$.

## C.3   EXPERIMENTAL DETAILS

The convolutional architecture that we use is identical to Wong & Kolter (2018), which consists of two convolutional layers with 16 and 32 filters of size $4 \times 4$ and stride 2, followed by a fully connected layer with 100 hidden units. The AT-$l_\infty$, AT-$l_2$, KW, MMR and MMR+AT training models are those presented in Croce et al. (2019a) and available at `https://github.com/max-andr/provable-robustness-max-linear-regions`. We trained the AT-$(l_1, l_2, l_\infty)$ performing for each batch of the 128 images the PGD-attack wrt the three norms (40 steps for MNIST and F-MNIST, 10 steps for GTS and CIFAR-10) and then training on the point realizing the maximal loss (the cross-entropy function is used), for 100 epochs. For all experiments with MMR-Universal we use batch size 128 and we train the models for 100 epochs. Moreover, we use Adam optimizer of Kingma & Ba (2014) with learning rate of $5 \times 10^{-4}$ for MNIST and F-MNIST, 0.001 for the other datasets. We also reduce the learning rate by a factor of 10 for the last 10 epochs. On CIFAR-10 dataset we apply random crops and random mirroring of the images as data augmentation.
For training we use MMR-Universal as in (9) with $k_B$ linearly (wrt the epoch) decreasing from 20% to 5% of the total number of hidden units of the network architecture. We also use a training schedule for $\lambda_p$ where we linearly increase it from $\lambda_p/10$ to $\lambda_p$ during the first 10 epochs. We employ both schemes since they increase the stability of training with MMR. In order to determine the best set of hyperparameters $\lambda_1$, $\lambda_\infty$, $\gamma_1$, and $\gamma_\infty$ of MMR, we perform a grid search over them for every dataset. In particular, we empirically found that the optimal values of $\gamma_p$ are usually between 1 and 2 times the $\epsilon_p$ used for the evaluation of the robust test error, while the values of $\lambda_p$ are more diverse across the different datasets. Specifically, for the models we reported in Table 4 the following values for the $(\lambda_1, \lambda_\infty)$ have been used: (3.0, 12.0) for MNIST, (3.0, 40.0) for F-MNIST, (3.0, 12.0) for GTS and (1.0, 6.0) for CIFAR-10.
In Tables 2 and 4, while the test error which is computed on the full test set, the statistics regarding upper and lower bounds on the robust test error are computed on the first 1000 points of the respective test sets. For the lower bounds we use the FAB-attack with the original parameters, 100 iterations and 10 restarts. For PGD we use also 100 iterations and 10 restarts: the directions for the update step are the sign of the gradient for $l_\infty$, the normalized gradient for $l_2$ and the normalized sparse gradient suggested by Tramèr & Boneh (2019) with sparsity level 1% for MNIST and F-MNIST, 10% for GTS and CIFAR-10. Finally we use the Liner Region Attack as in the original code. For MIP (Tjeng et al. (2019)) we use a timeout of 120s, that means if no guarantee is obtained by that time, the algorithm stops verifying that point.

Table 4: We report, for the different datasets and training schemes, the test error (TE) and lower (LB) and upper (UB) bounds on the robust test error (in percentage) wrt the $l_p$-norms at thresholds $\epsilon_p$, with $p = 1, 2, \infty$ (that is the largest test error possible if any perturbation of $l_p$-norm equal to $\epsilon_p$ is allowed). Moreover we show the $l_1 + l_2 + l_\infty$-UB, that is the upper bound on the robust error when the attacker is allowed to use the *union* of the three $l_p$-balls. The training schemes compared are plain training, adversarial training of Madry et al. (2018); Tramèr & Boneh (2019) (AT), robust training of Wong & Kolter (2018); Wong et al. (2018) (KW), MMR regularization of Croce et al. (2019a), MMR combined with AT (MMR+AT) and our MMR-Universal regularization. One can clearly see that our MMR-Universal models are the only ones which have non trivial upper bounds on the robust test error wrt all the considered norms.

**provable robustness against multiple perturbations**

| model | TE | $l_1$ LB | $l_1$ UB | $l_2$ LB | $l_2$ UB | $l_\infty$ LB | $l_\infty$ UB | $l_1+l_2+l_\infty$ LB | $l_1+l_2+l_\infty$ UB |
|---|---|---|---|---|---|---|---|---|---|
| **MNIST** | | $\epsilon_1 = 1$ | | $\epsilon_2 = 0.3$ | | $\epsilon_\infty = 0.1$ | | | |
| plain | 0.85 | 2.3 | 100 | 3.1 | 100 | 88.5 | 100 | 88.5 | 100 |
| AT-$l_\infty$ | 0.82 | 1.8 | 100 | 1.7 | 100 | 4.7 | 100 | 4.7 | 100 |
| AT-$l_2$ | 0.87 | 2.1 | 100 | 2.2 | 100 | 25.9 | 100 | 25.9 | 100 |
| AT-$(l_1, l_2, l_\infty)$ | 0.80 | 2.1 | 100 | 1.7 | 100 | 4.9 | 100 | 4.9 | 100 |
| KW-$l_\infty$ | 1.21 | 3.6 | 100 | 2.8 | 100 | 4.4 | 4.4 | 4.8 | 100 |
| KW-$l_2$ | 1.11 | 2.4 | 100 | 2.3 | 6.6 | 10.3 | 10.3 | 10.3 | 100 |
| MMR-$l_\infty$ | 1.65 | 10.0 | 100 | 5.2 | 100 | 6.0 | 6.0 | 10.4 | 100 |
| MMR-$l_2$ | 2.57 | 4.5 | 62.3 | 6.7 | 14.3 | 78.6 | 99.9 | 78.6 | 99.9 |
| MMR+AT-$l_\infty$ | 1.19 | 3.6 | 100 | 2.4 | 100 | 3.6 | 3.6 | 4.1 | 100 |
| MMR+AT-$l_2$ | 1.73 | 3.6 | 99.9 | 3.7 | 12.1 | 15.3 | 76.8 | 15.3 | 99.9 |
| MMR-Universal | 3.04 | 6.4 | 20.8 | 6.2 | 10.4 | 12.4 | 12.4 | 12.4 | **20.8** |
| **F-MNIST** | | $\epsilon_1 = 1$ | | $\epsilon_2 = 0.3$ | | $\epsilon_\infty = 0.1$ | | | |
| plain | 9.32 | 31.3 | 100 | 65.8 | 100 | 100 | 100 | 100 | 100 |
| AT-$l_\infty$ | 11.54 | 19.0 | 100 | 17.1 | 100 | 25.4 | 73.0 | 26.3 | 100 |
| AT-$l_2$ | 8.10 | 15.9 | 100 | 20.6 | 100 | 98.8 | 100 | 98.8 | 100 |
| AT-$(l_1, l_2, l_\infty)$ | 14.13 | 22.2 | 100 | 20.3 | 100 | 28.3 | 98.6 | 29.6 | 100 |
| KW-$l_\infty$ | 21.73 | 42.7 | 100 | 30.5 | 99.2 | 32.4 | 32.4 | 43.6 | 100 |
| KW-$l_2$ | 13.08 | 15.8 | 19.8 | 15.9 | 19.9 | 66.7 | 86.8 | 66.7 | 86.8 |
| MMR-$l_\infty$ | 14.51 | 28.5 | 100 | 23.5 | 100 | 33.2 | 33.6 | 36.7 | 100 |
| MMR-$l_2$ | 12.85 | 18.2 | 39.4 | 24.8 | 33.2 | 95.8 | 100 | 95.8 | 100 |
| MMR+AT-$l_\infty$ | 14.52 | 27.3 | 100 | 22.9 | 100 | 27.5 | 30.7 | 31.8 | 100 |
| MMR+AT-$l_2$ | 13.40 | 17.2 | 55.4 | 20.2 | 37.8 | 66.5 | 99.1 | 66.5 | 99.1 |
| MMR-Universal | 18.57 | 25.0 | 52.4 | 24.3 | 37.4 | 43.5 | 44.3 | 43.5 | **52.9** |
| **GTS** | | $\epsilon_1 = 3$ | | $\epsilon_2 = 0.2$ | | $\epsilon_\infty = {}^4/255$ | | | |
| plain | 6.77 | 60.5 | 100 | 38.4 | 99.3 | 71.1 | 98.4 | 71.5 | 100 |
| AT-$l_\infty$ | 6.83 | 64.0 | 100 | 24.9 | 99.2 | 31.7 | 82.3 | 64.0 | 100 |
| AT-$l_2$ | 8.76 | 44.0 | 100 | 27.2 | 98.4 | 58.9 | 97.1 | 59.0 | 100 |
| AT-$(l_1, l_2, l_\infty)$ | 8.80 | 41.8 | 100 | 24.0 | 93.7 | 41.2 | 79.4 | 45.2 | 100 |
| KW-$l_\infty$ | 15.57 | 87.8 | 100 | 41.1 | 77.7 | 36.1 | 36.6 | 87.8 | 100 |
| KW-$l_2$ | 14.35 | 46.5 | 100 | 30.8 | 35.3 | 57.0 | 63.0 | 57.6 | 100 |
| MMR-$l_\infty$ | 13.32 | 71.3 | 99.6 | 40.9 | 41.7 | 49.5 | 49.6 | 71.3 | 99.6 |
| MMR-$l_2$ | 14.21 | 54.6 | 80.4 | 36.3 | 36.6 | 62.3 | 63.6 | 62.6 | 80.9 |
| MMR+AT-$l_\infty$ | 14.89 | 82.8 | 100 | 39.9 | 44.7 | 38.3 | 38.4 | 82.8 | 100 |
| MMR+AT-$l_2$ | 15.34 | 49.4 | 84.3 | 33.2 | 33.8 | 57.2 | 60.2 | 58.1 | 84.8 |
| MMR-Universal | 15.98 | 49.7 | 51.5 | 34.3 | 34.6 | 47.0 | 47.0 | 51.6 | **52.4** |
| **CIFAR-10** | | $\epsilon_1 = 2$ | | $\epsilon_2 = 0.1$ | | $\epsilon_\infty = {}^2/255$ | | | |
| plain | 23.29 | 61.0 | 100 | 48.9 | 100 | 88.6 | 100 | 88.6 | 100 |
| AT-$l_\infty$ | 27.06 | 39.6 | 100 | 33.3 | 99.2 | 52.5 | 88.5 | 52.5 | 100 |
| AT-$l_2$ | 25.84 | 41.9 | 100 | 35.3 | 99.9 | 62.1 | 99.4 | 62.1 | 100 |
| AT-$(l_1, l_2, l_\infty)$ | 35.41 | 47.7 | 100 | 41.7 | 88.2 | 57.0 | 76.8 | 57.1 | 100 |
| KW-$l_\infty$ | 38.91 | 51.9 | 100 | 39.9 | 66.1 | 46.6 | 48.0 | 51.9 | 100 |
| KW-$l_2$ | 40.24 | 47.3 | 100 | 44.6 | 49.3 | 53.6 | 54.7 | 54.0 | 100 |
| MMR-$l_\infty$ | 34.61 | 54.1 | 100 | 42.3 | 68.4 | 57.7 | 61.0 | 58.7 | 100 |
| MMR-$l_2$ | 40.93 | 58.9 | 98.0 | 50.4 | 56.3 | 72.9 | 86.1 | 72.9 | 98.0 |
| MMR+AT-$l_\infty$ | 35.38 | 50.6 | 100 | 41.2 | 84.7 | 48.7 | 54.2 | 50.8 | 100 |
| MMR+AT-$l_2$ | 37.78 | 50.4 | 99.9 | 46.1 | 54.2 | 61.3 | 74.1 | 61.3 | 99.9 |
| MMR-Universal | 46.96 | 56.4 | 63.4 | 51.9 | 53.6 | 63.8 | 63.8 | 63.8 | **64.6** |

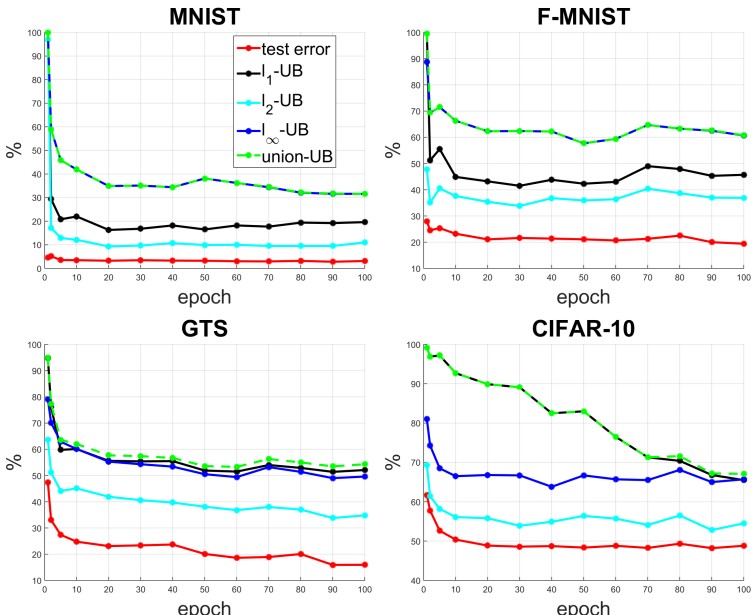

Figure 3: We show, for each dataset, the evolution of the test error (red), upper bound (UB) on the robust test error wrt $l_1$ (black), $l_2$ (cyan) and $l_\infty$ (blue) during training. Moreover, we report in green the upper bounds on the test error when the attacker is allowed to exploit the union of the three $l_p$-balls. The statistics on the robustness are computed at epoch $1, 2, 5, 10$ and then every 10 epochs on 1000 points with the method of Wong & Kolter (2018), using the models trained with MMR-Universal.

## C.4  EVOLUTION OF ROBUSTNESS DURING TRAINING

We show in Figure 3 the clean test error (red) and the upper bounds on the robust test error wrt $l_1$ (black), $l_2$ (cyan), $l_\infty$ (blue) and wrt the union of the three $l_p$-balls (green), evaluated at epoch $1, 2, 5, 10$ and then every 10 epochs (for each model we train for 100 epochs) for the models trained with our regularizer MMR-Universal. For each dataset used in Section 5 the test error is computed on the whole test set, while the upper bound on the robust test error is evaluated on the first 1000 points of the test set using the method introduced in Wong & Kolter (2018) (the thresholds $\epsilon_1, \epsilon_2, \epsilon_\infty$ are those provided in Table 1). Note that the statistics wrt $l_\infty$ are not evaluated additionally with the MIP formulation of Tjeng et al. (2019) as the results in the main paper which would improve the upper bounds wrt $l_\infty$.
For all the datasets the test error keeps decreasing across epochs. The values of all the upper bounds generally improve during training, showing the effectiveness of MMR-Universal.

## C.5  OTHER COMBINATIONS OF MMR AND AT

We here report the robustness obtained training with MMR-$l_p$+AT-$l_q$ with $p \neq q$ on MNIST. This means that MMR is used wrt $l_p$, while adversarial training wrt $l_q$. In particular we test $p, q \in \{1, \infty\}$. In Table 5 we report the test error (TE), lower (LB) and upper bounds (UB) on the robust test error for such model, evaluated wrt $l_1$, $l_2$, $l_\infty$ and $l_1 + l_2 + l_\infty$ as done in Section 5. It is clear that training with MMR wrt a single norm does not suffice to get provable guarantees in all the other norms, despite the addition of adversarial training. In fact, for both the models analysed the UB equals 100% for at least one norm. Note that the statistics wrt $l_\infty$ in the plots do not include the results of the MIP formulation of Tjeng et al. (2019).

Table 5: Robustness of other combinations of MMR and AT.

| | | $l_1$ | | $l_2$ | | $l_\infty$ | | $l_1 + l_2 + l_\infty$ | |
|---|---|---|---|---|---|---|---|---|---|
| *model* | TE | LB | UB | LB | UB | LB | UB | LB | UB |
| MMR-$l_1$+AT-$l_\infty$ | 0.99 | 2.5 | 15.7 | 2.6 | 25.4 | 9.4 | 100 | 9.4 | 100 |
| MMR-$l_\infty$+AT-$l_1$ | 1.43 | 2.7 | 100 | 2.3 | 100 | 5.6 | 30.2 | 5.6 | 100 |

### C.6 Larger models

We trained models with MMR-Universal also on the "Large" architecture from Wong et al. (2018), but we could not achieve a significant improvement compared to the smaller networks reported in the main paper. Note that the verification becomes the more expensive the larger the network is. Thus for the results in Table 6 we use only the method of Wong & Kolter (2018) to compute the UB on the robust test error (we do not additionally use Tjeng et al. (2019) for the statistics relative to the $l_\infty$-robustness which yields tighter upper bounds), which explains why the results are seemingly worse than those in Table 2.

Table 6: MMR-Universal models trained on the "Large" architecture from Wong et al. (2018) with the same $\epsilon_p$ as in the experiments in Section 5. * For the $l_\infty$-robustness in this case the method of Tjeng et al. (2019) is not used.

**MMR-Universal models with larger architecture**

| | | $l_1$ | | $l_2$ | | $l_\infty$ | | $l_1 + l_2 + l_\infty$ | |
|---|---|---|---|---|---|---|---|---|---|
| *model* | TE | LB | UB | LB | UB | LB | UB | LB | UB |
| MNIST | 2.20 | 4.6 | 11.2 | 4.2 | 6.9 | 8.8 | 48.8* | 8.8 | 48.8 |
| F-MNIST | 19.20 | 29.8 | 47.6 | 25.5 | 34.2 | 43.6 | 64.4* | 43.7 | 64.5 |
| GTS | 17.40 | 45.8 | 54.8 | 32.1 | 36.9 | 47.5 | 58.3* | 49.8 | 60.1 |
| CIFAR-10 | 45.09 | 48.6 | 63.3 | 46.0 | 48.9 | 57.1 | 69.6* | 57.1 | 69.9 |

