# OpenReview forum: "Provable robustness against all adversarial $l_p$-perturbations for $p\geq 1$"
_ICLR.cc/2020/Conference — Accept (Poster)_

### Official Review · AnonReviewer1 · 2019-10-23
**Official Blind Review #1**

**Rating:** 6

**Review:**

Overview:

The paper is dedicated to developing a regularization scheme for the provably robust model. The author proposes the MMR-Universal regularizer for ReLU based networks. It enforces l1 and l infinity robustness and leads to be provably robust with any lp norm attack for p larger than and equal to one.

Strength Bullets:

1. Using convex hull to enforce robustness, it is very reasonable and straightforward which highly aligns our intuition.
2. The author provides detailed and conniving derivations and proof. And the results in table 2 do achieve the state-of-the-art provable robustness.
3. It is the first robust regularizer that is able to provide non-trivial robustness guarantees for multiple lp balls. The lp balls don't contain any other. It also is one of the most straightforward methods among all potential similar methods.

Weakness Bullets:

1. I am very curious about the landscape or decision boundary analysis and visualization. For the author's MMR-Universal regularization, it should give the model a very "good" decision boundary which has clear marginal between any two categories. In my opinion, it is necessary to evidence to convince more readers.
2. Try a few more classical CNN architectures.

Recommendation:

Due to the logical derivations and supportive experiment results, this is a weak accept.

**Experience Assessment:**

I have read many papers in this area.

**Review Assessment: Checking Correctness Of Derivations And Theory:**

I assessed the sensibility of the derivations and theory.

**Review Assessment: Checking Correctness Of Experiments:**

I carefully checked the experiments.

**Review Assessment: Thoroughness In Paper Reading:**

I read the paper at least twice and used my best judgement in assessing the paper.

---

> ### Author Response · Authors · 2019-11-14
> **Answer to reviewer 1**
>
> We thank the reviewer for the encouraging comments. We address below the questions:
>
> 1) "``I am very curious about the landscape or decision boundary analysis and visualization"
>
> In general, MMR-Universal significantly expands the regions where the classifier is affine, similarly to what MMR does. The difference is that MMR-Universal enlarges the linear regions so that all the $l_p$-balls for $p\geq 1$ are contained. Since the decision boundary inside each linear region consists in a hyperplane and the linear regions are especially wide, the decision boundary is likely to be the union of portions of a few hyperplanes. The problem however is that this effect is in $2$ dimensions relatively limited as the differences between $l_p$-balls become much more pronounced  with increasing input dimension $d$.  In this short amount of time we could not yet come up  with a good example in $2$ dimensions resp. with an appropriate projection of the case in higher dimensions.
> We agree with the reviewer that such an illustration will further provide intuition and is thus helpful and we will try hard to come up with this for the final version.
>
> 2) "``Try a few more classical CNN architectures"
>
> We will train for the final version the usual LeNet architecture and the large network from (Wong et al, 2018), as it is a standard choice for provable robustness, which is roughly 4 times larger than our current one. Moreover, we will extend the implementation of our method to handle residual connections, which do not make any difference to the algorithm from a theoretical point of view. At the moment we are running low on GPU resources due to the upcoming CVPR deadline - we are sorry that we have to postpone the additional results for larger models to the final version.

---

### Official Review · AnonReviewer2 · 2019-10-23
**Official Blind Review #2**

**Rating:** 8

**Review:**

Summary of the paper's contributions:

This paper proves a result on the l_p robustness (p \neq 1, \infty) of a piecewise affine classifier in terms of its l_1 and l_\infty robustness. This result is based on the insight that a guarantee for l_1 and l_\infty robustness of a piecewise affine classifier also guarantees robustness within the convex hull of the union of the l_1 and l_\infty balls. The paper then proposes a regularization scheme called MMR-Universal for a ReLU neural network that simultaneously trains for l_1 and l_\infty robustness. This scheme is based on maximizing the linear regions of the network as in Croce et al., AISTATS 2019. Using the main result of the paper, it is implied that the proposed regularization scheme also enforces l_p robustness for any p \geq 1.

Main comments: (1) The paper provides an interesting result that guarantees l_p robustness for a piecewise affine classifier based on only l_1 and l_\infty robustness. (2) The experiments show that the proposed regularization scheme is indeed effective in simultaneously guaranteeing robustness with respect to all l_p balls.

Detailed comments:

- The proposed regularization scheme does not come with any explicit robustness guarantee. Is it possible to show that a model that minimizes the regularized loss is guaranteed to be l_p robust for some radius? In Appendix C.3, it is mentioned that the best values for \gamma_p were empirically found to be 1-2 times the desired \epsilon_p robustness. Could this be formalized in theory?

- From Figure 2, it seems that there is an optimal ratio of l_1 to l_\infty robustness for which the l_2 guarantee is maximized. It would be interesting to see what this optimal ratio is for maximizing the l_p guarantee, as a function of dimension d and p.

- In the experiments, the MMR-Universal scheme is compared with MMR+AT-l_2 and MMR+AT-l_\infty. It would be interesting to compare it with MMR-l_1+AT-l_\infty or with MMR-l_\infty+AT-l_1.

- In eq(1), is c intended to be the true label at x?

- In Figure 2, the red curves are missing in the first and third plots.

**Experience Assessment:**

I have read many papers in this area.

**Review Assessment: Checking Correctness Of Derivations And Theory:**

I assessed the sensibility of the derivations and theory.

**Review Assessment: Checking Correctness Of Experiments:**

I assessed the sensibility of the experiments.

**Review Assessment: Thoroughness In Paper Reading:**

I read the paper thoroughly.

---

> ### Author Response · Authors · 2019-11-14
> **Answer to Reviewer 2**
>
> We thank the reviewer for the encouraging comments. We address the raised questions individually.
>
> 1) "The proposed regularization scheme does not come with any explicit robustness guarantee. Is it possible to show that a model that minimizes the regularized loss is guaranteed to be $l_p$ robust for some radius?"
>
> If the MMR-Universal regularization (stated in page 6) is successfully minimized during training and reaches zero for a particular training point, then for this point we have achieved robustness of $\gamma_1$ wrt $l_1$ and $\gamma_\infty$ of $l_\infty$, since no hyperplane, either region or decision boundary, is closer than those thresholds to any of the points of the training set (please see Theorem 2.1).
> Theorem 4.1 provides then for any $p\in(1, \infty)$ in Equation (7) the radius of the $l_p$-ball on which the decision does not change and is thus robust. In practice, on the test points some hyperplanes happen to be closer than the given thresholds even after training, so that the guarantees from Theorem 4.1 are smaller than the desired ones. However, since MMR-Universal expands also neighboring linear regions (where there are no training points), we can certify with the mixed-integer programming certification of (Tjeng et al, 2019) or the approach of (Wong and Kolter, 2018) $l_p$-balls with larger radii, achieving good provable robustness. As for all current certification methods we observe that the enforced robustness on the training set generalizes to the test set.
>
> 2) "In Appendix C.3, it is mentioned that the best values for $\gamma_p$ were empirically found to be 1-2 times larger than the desired $\epsilon_p$ robustness. Could this be formalized in theory?"
>
> The effectiveness of larger $\gamma$ is an empirical observation, see also e.g. (Gowal et al, 2018). In practice, we use $\gamma > \epsilon$ in order to counter the fact that potentially the training is able to push the hyperplanes only partially, and not up to the desired threshold. It is an interesting question if one can
> back up this procedure with theoretical arguments.
>
> 3) "From Figure 2, it seems that there is an optimal ratio of $l_1$ to $l_\infty$ robustness for which the $l_2$ guarantee is maximized. It would be interesting to see what this optimal ratio is for maximizing the $l_p$ guarantee, as a function of dimension d and p."
>
> Thanks for this interesting question. We added in the Appendix A.3 the derivation of the optimal ratio as a function of $d$ for $p=2$, as well as the corresponding maximal improvement  of the bounds given by the convex hull over those obtained considering only the union of the $l_1$ and $l_\infty$ balls. It turns out that the gain of the $l_2$-robustness guarantee of the convex hull  over to the union of the $l_1$-and $l_\infty$-ball increases with dimension $d$ as $d^\frac{1}{4}$. For $p\neq 2$, it needs to be approximated numerically.
>
> 4) "In the experiments, the MMR-Universal scheme is compared with MMR+AT-$l_2$ and MMR+AT-$l_\infty$. It would be interesting to compare it with MMR-$l_1$+AT-$l_\infty$ or with MMR-$l_\infty$+AT-$l_1$."
>
> We have started the training and the evaluation of such models of MNIST and report them in Appendix C.5. We will add the other datasets in the final version. The preliminary analysis on MNIST suggests that the models have good provable robustness only in the norm for which MMR is used. Moreover, the model trained with MMR-$l_1$+AT-$l_\infty$ shows non-trivial provable robustness also wrt $l_2$ (as it happens sometimes in Table 4), but comes with no guarantees wrt $l_\infty$. In particular, we achieve no non-trivial robustness for the union of the $l_1$, $l_2$, $l_\infty$-balls. This stresses again that enforcing provable robustness wrt
> $l_1$ and $l_\infty$ as in MMR-Universal is the key for provable robustness wrt intermediate $l_p$-balls.
>
> 5) "In eq(1), is c intended to be the true label at x?"
>
> Yes, thanks for noticing this. We have fixed this.
>
> 6) "In Figure 2, the red curves are missing in the first and third plots."
>
> The red curve is hidden by the green one, as they almost everywhere overlap and they are difficult to distinguish. The red curves can be seen for values of $\norm{x}_1$ close to 1, zooming into the plots. We have added a note on this in the caption.

---

> > ### Author Response · Authors · 2019-11-14
> > **Typo**
> >
> > Sorry, it should $\|x\|_1$ in the answer to 6)

---

### Official Review · AnonReviewer3 · 2019-10-27
**Official Blind Review #3**

**Rating:** 6

**Review:**

Summary:
The author proposed MMR regularization for the provable robustness of union of l-1 and l-infty balls, which is robust to any l-p norm for p>=1.

Strengths:
1. The paper is well organized.
2. The theoretical part is completed and experiments are done on MNIST, Fashion-MNIST, GTS, CIFAR-10.
3. The proposed method presents a significant improvement over SOTA.

Weakness:
1. The author should provide more empirical analysis on the MMR regularization, like how it changes during the training process.
2. The captions of the table and figure are too small.


**Experience Assessment:**

I have read many papers in this area.

**Review Assessment: Checking Correctness Of Derivations And Theory:**

I assessed the sensibility of the derivations and theory.

**Review Assessment: Checking Correctness Of Experiments:**

I assessed the sensibility of the experiments.

**Review Assessment: Thoroughness In Paper Reading:**

I made a quick assessment of this paper.

---

> ### Author Response · Authors · 2019-11-14
> **Answer to Reviewer 3**
>
> We thank the reviewer for the detailed comments. We have addressed the suggested changes:
>
> 1) "The author should provide more empirical analysis on the MMR regularization, like how it changes during the training process."
>
> We have added in the Appendix C.4 plots showing the evolution of test error and upper bounds on the robust $l_p$-test error for $p \in \{1,2,\infty\}$  evaluated using the approach of (Wong and Kolter, 2018) together with an upper bound on the robust test error over the union of these $l_p$-balls  during training using our regularizer MMR-Universal for all four datasets. For all datasets we see improving test error as well as robustness with increasing number of epochs. Please note that the reported results differ from the ones reported in the main paper as we had to retrain the models to do the evaluation during training but most importantly due to the fact that we do not use the expensive mixed-integer programming of approach (Tjeng et al, 2019) for $l_\infty$ which would improve the $l_\infty$ upper bounds on the robustness quite significantly but is very slow so that a full evaluation every 10 epochs or less is quite time consuming (please note that we are running a bit low on computational resources due to the upcoming CVPR deadline). We will include the evaluation with MIP in the final version.
>
>
> 2) "The captions of the table and figure are too small."
>
> We have increased the size of the captions for better readability.

---

### Decision · Program_Chairs · 2019-12-19

**Decision:**

Accept (Poster)

**Comment:**

This paper extends the degree to which ReLU networks can be provably resistant to a broader class of adversarial attacks using a MMR-Universal regularization scheme.  In particular, the first provably robust model in terms of lp norm perturbations is developed, where robustness holds with respect to *any* p greater than or equal to one (as opposed to prior work that may only apply to specific lp-norm perturbations).

While I support accepting this paper based on the strong reviews and significant technical contribution, one potential drawback is the lack of empirical tests with a broader cohort of representative CNN architectures (as pointed out by R1).  In this regard, the rebuttal promises that additional experiments with larger models will be added in the future to the final version, but obviously such results cannot be used to evaluate performance at this time.